# Physiological and Antioxidant Response to Different Water Deficit Regimes of Flag Leaves and Ears of Wheat Grown under Combined Elevated CO_2_ and High Temperature

**DOI:** 10.3390/plants11182384

**Published:** 2022-09-13

**Authors:** Ouardia Bendou, Ismael Gutiérrez-Fernández, Emilio L. Marcos-Barbero, Nara Bueno-Ramos, Jon Miranda-Apodaca, Ana I. González-Hernández, Rosa Morcuende, Juan B. Arellano

**Affiliations:** 1Institute of Natural Resources and Agrobiology of Salamanca (IRNASA), Consejo Superior de Investigaciones Científicas (CSIC), 37008 Salamanca, Spain; 2Department of Plant Biology and Ecology, Faculty of Science and Technology, University of the Basque Country (UPV/EHU), Apdo. 644, 48080 Bilbao, Spain

**Keywords:** antioxidant, ascorbate-glutathione cycle, anisohydric behavior, ear, elevated CO_2_, flag leaf, high temperature, leaf photosynthesis, water deficit, wheat

## Abstract

*Triticum aestivum* L. cv. Gazul is a spring wheat widely cultivated in Castilla y León (Spain). Potted plants were grown in a scenario emulating the climate change environmental conditions expected by the end of this century, i.e., with elevated CO_2_ and high temperature under two water deficit regimes: long (LWD) and terminal (TWD). Changes in biomass and morphology, the content of proline (Pro), ascorbate (AsA) and glutathione (GSH), and enzymatic antioxidant activities were analyzed in flag leaves and ears. Additionally, leaf gas exchange was measured. LWD caused a decrease in biomass and AsA content but an increase in Pro content and catalase and GSH reductase activities in flag leaves, whereas TWD produced no significant changes. Photosynthesis was enhanced under both water deficit regimes. Increase in superoxide dismutase activity and Pro content was only observed in ears under TWD. The lack of a more acute effect of LWD and TWD on both organs was attributed to the ROS relieving effect of elevated CO_2_. Gazul acted as a drought tolerant variety with anisohydric behavior. A multifactorial analysis showed better adaptation of ears to water deficit than flag leaves, underlining the importance of this finding for breeding programs to improve grain yield under future climate change.

## 1. Introduction

The latest Intergovernmental Panel on Climate Change report showed that the atmospheric CO_2_ concentration increased from 300 ppm in 1900 to 410 ppm in 2019, and it is expected to reach 867–1135 ppm by 2100 (based on the share socioeconomic pathways SSP3-7.0, SSP5-5.8), causing a concomitant increase in the average global temperature by 1.3 to 1.9 °C by 2040 and 3.3 °C to 5.7 °C by the end of this century (SSP5-8.5) [1]. The warming climate is also expected to lead increases in soil water evaporation and the capacity of air to hold water, which, in turn, may cause prolonged dry episodes, exposing more land to drought [2]. Besides soil water evaporation, the dry episodes would also be brought about by the absence, or low intensity, of rainfalls and the deviation of typical seasonal patterns, which are predicted to increase in the future [3,4]. Undoubtedly, global climate change will negatively affect agriculture, setting a challenge for sustainable agricultural production in a world where the human population may reach nine billion people by 2050 and the demand for food production must increase accordingly [5,6,7].

Wheat, maize and rice are among the three most important food crops worldwide. The former is widely cultivated in temperate regions all over the world for use as human food and livestock feed because of its nutritional value as a major source of energy, protein, dietary fiber and micronutrients [8,9,10,11]. According to the Food and Agriculture Organization of the United Nations, worldwide wheat production was 769.6 Mt in 2021, which was consumed by 35% of the population. Despite the need for a sustainable increase in crop production, global crop losses due to drought events were assessed to be 8% from 1983 to 2009, and the trend in future wheat production was projected to decline by 2.5% due to global climate change [5,12]. Remarkably, wheat production will drop even more severely than expected in 2022 to about 13% below the output of the last two years due to additional factors such increasing energy rates and shortages of fertilizers and improved seeds [13].

Because of the deleterious impact that elevated temperature and drought episodes associated with the increase in atmospheric CO_2_ concentrations has on agriculture, there is an urgent need to better understand plant responses to these environmental factors, with the aim of selecting climate-resilient crops that can sustainably provide food for an ever-growing worldwide population. Although plant responses to these environmental factors are individually well established, when combined, analyses are not without complexities. In the first instance, elevated CO_2_ availability stimulates the net CO_2_ assimilation rate (*A*) in C_3_ plants (including wheat) in the short term and, consequently, increases the carbohydrate production, growth and development [14,15]. CO_2_ enrichment produces a decrease in the stomatal conductance to water vapor (*g*_s_) and transpiration rate (*E*), causing an additional improvement in water use efficiency (WUE) [16]. However, most C_3_ plants fail to sustain the *A* stimulation in the long term and, eventually, prolonged exposure to elevated CO_2_ leads to a loss in photosynthesis capacity [17,18,19]. This phenomenon, named photosynthetic acclimation to elevated CO_2_, is triggered by different factors, but carbohydrate repression of photosynthetic genes is considered to play a crucial role [15,20]. The loss in photosynthesis capacity is accompanied by a noticeable decline in ribulose-1,5-bisphosphate carboxylase-oxygenase (Rubisco) content and activity [21,22,23] due to insufficient plant sink strength [24] or lower plant nitrogen content [25,26,27]. The changes in carbon and nitrogen metabolism under elevated CO_2_ concentration are also accompanied by changes in plant respiration and antioxidant protection processes against reactive oxygen species (ROS) generation, which ameliorate plant response to oxidative stress [17].

Secondly, wheat is highly sensitive to high temperatures; an increase in 1 °C can decrease its production by 3 to 10% [7] and its grain quality [28,29]. This is a consequence of the detrimental effect of temperature on the photosynthetic machinery and the enhancement of the photorespiration rate. Photosystem II (PSII) is one of the most heat-labile components of the photosynthetic apparatus, although enzymes of the Calvin-Benson cycle, such as Rubisco, and Rubisco activase, are also characterized by their prompt inactivation, even at moderate-high temperatures [30]. Thus, high temperatures limit the production of photo-assimilates, increase the electron flow toward the oxygenation of the ribulose-1,5-bisphosphate and ultimately reduce plant growth [31,32]. In addition, temperature-induced oxidative stress disrupts thylakoids and the plasma membrane and boosts protein denaturation and chlorophyll (Chl) degradation [16,28,32,33].

Third, drought stress is considered to be one of the major environmental factors shaping the geographic distribution of plants. It affects plants at different growth stages, from germination to maturity, causing a reduction in crop growth and productivity [34,35,36] that can range from about 1% to 30% under mild drought stress at post-anthesis up to 92% under prolonged drought at flowering and grain formation [7,37]. Photosynthesis is one of the primary biological processes affected by drought [38]. Under limited water uptake from the soil, plant hormone regulation, especially by abscisic acid (ABA), drives stomatal closure through the production of key signaling molecules such as H_2_O_2_ to reduce *E* and so plant water content loss [39]. A decrease in *g*_s_ also lowers the intercellular concentration of CO_2_ (*C_i_*), reduces *A* and disrupts the primary electron transport reactions in thylakoids [40,41]. The oxygenation of ribulose-1,5-bisphosphate by Rubisco is particularly enhanced and contributes to ROS overproduction, together with the reduction of oxygen to superoxide radical (O_2_^●^^−^) through the Mehler reaction [41,42,43]. Under these conditions, an increase in enzymatic antioxidant activity—especially in catalase (CAT), superoxide dismutase (SOD), ascorbate peroxidase (APX) and glutathione reductase (GR)—is observed in parallel with an increase in the content of proline (Pro), which acts as an osmoprotectant and ROS scavenger [44,45]. In addition, the ratio between reduced and oxidized ascorbate (AsA and DHA) in flag leaves and ears significantly decreases under drought conditions, while the ratio between reduced and oxidized glutathione (GSH and GSSG) increases under the same conditions [44,46,47].

In the aforementioned studies, plant responses to elevated atmospheric CO_2_ concentration, high temperature and drought were treated discreetly, even though they occur simultaneously, and even interact with each other, in nature. With regard to CO_2_ enriched environments, there is a growing body of evidence showing that elevated CO_2_ can mitigate the negative effects of other abiotic stress factors such as drought and high temperature or the interacting effect of both factors [17,19,27,48,49,50]. Elevated CO_2_ has been shown to alleviate the negative effect of drought stress by enhancing *A*, reducing oxidative stress and improving WUE [48,51,52]. Similarly, it was shown that elevated CO_2_ partly attenuated the decrease in photosynthesis and enzymatic antioxidant activity triggered by high temperature [53] and the repression of genes for photosynthesis and other primary and secondary metabolic processes [17]. The combination of heat waves and drought enhanced oxidative stress and brought about a reduction of *A* and *g*_s_ together with a decrease in biomass and grain yield production in wheat plants [54,55,56]. However, the elevated CO_2_ was shown to mitigate the effects of the former two factors combined due to the up-regulation of the antioxidant defense metabolism and an enhancement of photosynthesis and biomass, which were accompanied by an improvement in the water status, nitrogen use efficiency and cellular membrane stability, although there was no improvement in grain yield and quality [55,56,57].

In the present study, special emphasis was placed upon two organs, i.e., the flag leaf and ear, of the spring wheat cultivar Gazul, exposed to two different water deficit regimes—defined here as long and terminal water deficit regimes (LWD and TWD)—under growing conditions in which the CO_2_ concentration was set at 900 ppm and the temperature was 4 °C above the current average temperatures during the life cycle of spring wheat in the province of Salamanca (Spain). Organ biomass and morphology, antioxidant and osmoprotectant metabolite content and enzymatic antioxidant activity, including that of enzymes belonging to the AsA-GSH cycle, were evaluated, together with *A* of flag leaves at the anthesis growth stage. Multiple factor analysis (MFA) revealed that the two organs responded to the two water deficit regimes differently, and that the status of one organ did not completely reflect the status of the other.

## 2. Results

### 2.1. Multiple Factor Analysis

A series of MFA was initially performed to explore in depth the correlation between the biomass, morphology, metabolite content and enzymatic antioxidant activity of the flag leaves and ears of potted wheat plants under control and water deficit conditions. Two categorical factors were considered: *ORGAN* (flag leaves and ears) and *TREATMENT* (control, LWD and TWD). The continuous variables were divided into four groups: *Biomass* [Fresh weight (FW), dry weight (DW) and moisture content percentage (MC%)], *Morphology* [Area and DW per unit of area (DMA)], *Metabolites* (Pro, AsA and GSH) and *Antioxidant enzymes* [SOD, CAT, APX, dehydroascorbate reductase (DHAR), monodehydroascorbate reductase (MDHAR) and GR]. In the first exploratory analysis, the MFA showed that the first two dimensions comprised 72.40% of the total variance (Appendix A). *ORGAN* was the main factor explaining the data variability and was strongly correlated with the first dimension (0.95). In contrast, *TREATMENT* had a lower weight and showed better correlation with the second dimension (0.37) (Appendix A). The individuals-MFA plot showed that the individuals of both organs were clearly separated along the first dimension, while those of control and TWD-treated plants were separated from the LWD-treated plants along the second dimension (Appendix A). When both organs were compared, the ears were found to be particularly associated with higher values for FW, DW, Area and DMA, together with enhanced SOD, DHAR and APX activity (correlations between 0.7–0.97) (Appendix A). On the other hand, LWD-treated plants were associated with lower values for MC% and AsA content but enhanced activity for CAT and GR and higher Pro content when compared to control and TWD-treated plants (Appendix A). These associations were confirmed by the correlogram based on Spearman’s correlation coefficient (*r_s_*), showing positive correlation between MC% and AsA content (0.51), as well as between GR and Pro content (0.67), but negative correlations of MC% and AsA content with CAT, GR and Pro content (from −0.26 to −0.64) (Appendix A).

Based on the different characteristics displayed by the individuals of flag leaves and ears, the data were divided into two sets, each corresponding with one organ, for further exploration (Appendix A). Accordingly, two new MFA were performed in which the continuous variable groups remained as described above. First, MFA with the data belonging to the flag leaves showed that the first dimension comprised 64.50% of the total variance (Figure 1A), and that the *TREATMENT* factor was strongly correlated with the first dimension (0.87). The continuous variable groups with the highest contribution to, and strongest correlation with, the first dimension were *Biomass* (25.95, 0.94) and *Morphology* (25.94, 0.94), closely followed by *Metabolites* (25.56, 0.93). *Antioxidant enzymes* (22.55, 0.82) was the variable with the lowest contribution to the first dimension (Figure 1A). Regarding the individual variables (Figure 1B), those that contributed the most to the first dimension were Area (13.28), DMA (12.66), AsA content (11.35) and MC% (9.15). The position of the group of individuals belonging to the LWD treatment were distinctly separated from the control group position in the individuals-MFA plot, while the position of the group of individuals of the TWD treatment overlapped the control group (Figure 1C). The LWD treatment was found to be associated with higher DMA, Pro content and GR, CAT and APX activities, but lower FW, DW, MC%, Area, AsA content and DHAR activity (Figure 1B,C). In this respect, positive correlations were found for GR with CAT (0.79) and Pro content (0.93), as well as among FW, DW, MC%, Area and AsA content (0.54–0.99). Furthermore, negative correlations were found for CAT, GR and Pro content with AsA content, FW, DW, MC% and Area (from −0.48 to −0.80) (Appendix A).

Second, our analysis of the data of the ears showed that the first dimension represented 35.59% of the total variance. In addition, the *TREATMENT* factor was strongly correlated with the first dimension (0.82) (Figure 1D). The variable groups with the highest contribution to, and strongest correlation with, the first dimension were *Biomass* (37.59, 0.94) and *Morphology* (36.33, 0.91). They were followed by *Metabolites* (21.12, 0.53), but *Antioxidant enzymes* (4.96, 0.12) was weakly correlated with the first dimension. Regarding the individual variables, the ones that contributed highest to the first dimension were DMA (23.62), followed by GSH (13.93), MC% (12.92), Area (12.72), DW (12.61) and FW (12.06) (Figure 1E). The individuals-MFA plot also disclosed that the individuals of both groups of LWD- and TWD-treated plants overlapped and were separated from the individuals of the group of control plants. The projection of the continuous variables to the individuals-MFA plot showed that both LWD and TWD treatments were associated with higher MC%, Pro content and MDHAR, GR, APX and CAT activities but lower FW, DW, DMA, Area, and GSH content in the ear (Figure 1E,F). Positive and significative correlations were found among FW, DW, Area and DMA (0.58–0.94), and negative correlations of MC% with these parameters (from −0.52 to −0.96) (Appendix A).

The *ORGAN* factor had the greatest contribution to the data variability of the present study. Additionally, it was concluded that the effect of the *TREATMENT* factor could be better assessed if the data were divided into two groups. The following sections highlight the mains effects of the water deficit treatments on the measured parameters of each organ individually.

### 2.2. Biomass and Morphology

Drought stress is known to exert detrimental effects on plant organ biomass and morphology. These effects were observed in the present study, although the effects varied notably depending on the water deficit regime applied. The FW and DW of both the flag leaf and ear organs were especially affected by the LWD treatment (Figure 2A,B). The FW and DW biomass of the flag leaf of the plants exposed to the LWD treatment decreased with statistical significance when compared to the control conditions. However, the TWD treatment led to a slight decrease in the FW and DW of the flag leaves without reaching statistical significance (Figure 2A). Similar trends were observed for the FW and DW of the ears when the LWD treatment was compared to control conditions. The FW of the ears belonging to the TWD-treated plants presented intermediate values, i.e., between those of control and LWD-treated plants, while the DW of the ears of the TWD-treated plants decreased with statistical significance (Figure 2B). These results indicated that the LWD treatment caused the most detrimental effect on the FW and DW biomass of the flag leaf and ear organs under the designed experimental conditions. The MC% of the flag leaves of LWD-treated plants decreased with statistical significance compared to those of the control and TWD-treated plants, whereas the MC% of the ears of both LWD- and TWD-treated plants was statistically significantly higher than that of control plants (Figure 1A,B).

Together with the FW and DW biomass, the green areas of the flag leaves and ears were measured (Figure 2C). Our statistical analysis indicated that the LWD treatment had a severe negative impact on the green area of the flag leaves, which were notably smaller than those of the control plants, although intriguingly, they were thicker and had higher DMA (Figure 2D). In contrast to the flag leaves, the green area of the ears of the plants exposed to both LWD and TWD treatments did not significantly change compared to the control ones (Figure 2C). The DMA of the ears was smaller in plants under drought stress, regardless of the water deficit regime (Figure 2D).

### 2.3. Chlorophyll and Photosynthesis Status

Changes in the leaf Chl content caused by environmental stresses are characterized by bringing together alterations in *A*. Firstly, in contrast to expectations, the content of Chl measured on the attached flag leaves was unaffected by the water deficit regimes, and the values in Dualex units were similar to those of the control plants (Figure 3A).

Second, *A* at 900 ppm of CO_2_ (growth chamber conditions) was higher in both LWD and TWD treatments and statistically significant when compared to *A* under water control conditions (Figure 3B). The hyperbolic fitting of *A* in the range between 60–2000 ppm of CO_2_ revealed that the maximum carboxylation rate by Rubisco (*V_cmax_*) and maximum electron transport rate (*J_max_*) were statistically significantly higher in both water deficit regimes (Table 1). These two photosynthetic parameters were accompanied by a concomitant increase in *g*_s_ and *E*, although the latter two parameters did not reach statistical significance. When the WUE was computed as the ratio between *A* and *E*, no statistically significant difference between treatments was found (Figure 3C). The respiration rate in the light (*R_d_*) of the flag leaves was statistically significantly higher under control conditions compared to the two water deficit treatments (Table 1).

### 2.4. Accumulation of Metabolites with Osmoprotectant and Antioxidant Roles

The lack of an obvious detrimental effect on the photosynthesis activity of the flag leaves under both water deficit regimes led us to question whether the severity of the treatments had reached a triggering threshold that was high enough to induce the biosynthesis of metabolite markers of drought stress. Both water deficit treatments affected the Pro content in flag leaves and ears, but in different manner (Figure 4A).

While a statistically significant increase in the Pro content was observed in the flag leaves of LWD-treated plants compared to control plants, the Pro content remained unaltered in TWD-treated plants, a result that differed from the changes in the Pro content in the ears, in which the more noticeable increase in the Pro content with statistical significance was observed in TWD-treated plants. There was also a slight increase, but without statistical significance, in the Pro content in the ears of LWD-treated plants compared to that in the control plants.

The pattern of changes in the Pro content in contrast to that observed for the total content of AsA, which was found to be low when the Pro content was high. The LWD treatment induced a statistically significant decrease in the AsA content of flag leaves, but it did not change in the TWD treatment relative to the control conditions (Figure 4B). Although there was no statistically significant difference in the AsA content in the ears, a slight decline was found in TWD-treated plants. Attempts to measure DHA in both organs were conducted; however, the content of DHA was concluded to be ≤5% of the total content of AsA plus DHA, i.e., below the experimental error of the analytical method. Regarding GSH, both LWD and TWD treatments caused a decrease in the total content of GSH with statistical significance in the ears, but not in the flag leaves (Figure 4C).

### 2.5. Superoxide Dismutase and Catalase

Changes in the activity of SOD and CAT were monitored in flag leaves and ears under the two water deficit treatments (Figure 5). The SOD activity marginally decreased in the flag leaves under both LWD and TWD treatments, although the loss of activity was not statistically significant when compared to that in flag leaves of control plants (Figure 5A). The SOD activity only showed an increase in the ears of TWD-treated plants with statistical significance when compared to LWD-treated plants, but not when compared to the control conditions. In contrast to the SOD activity, the CAT activity was enhanced in both organs under the two water deficit regimes, although the changes were only statistically significant in the flag leaves of the LWD-treated plants (Figure 5B). Overall, the results for both CAT and SOD activities suggested that the enhancement of ROS production was rather mild or moderate in the flag leaves and ears under both water regimes.

### 2.6. Ascorbate Glutathione Cycle

The antioxidant activities of APX, DHAR and DHAR, belonging to the AsA-GSH cycle, were rather similar between treatments (Figure 6). Statistically significant differences could only be found between organs in some enzyme activities, regardless of the water deficit regime applied (See Appendix A). In particular, the APX activity was statistically significantly higher in the ears than in the flag leaves, a result that went in parallel with the values obtained for the DHAR activity in both organs. In contrast, there were no statistically significant differences in MDHAR activity between organs or treatments. Among the enzymatic antioxidant activities measured within the AsA-GSH cycle, only the GR activity exhibited a statistically significant increase in the flag leaf of the LWD-treated when compared to the TWD-treated and control plants. However, the GR activity showed no significant variation in the ears (Figure 6D).

## 3. Discussion

*Triticum aestivum* L. cv. Gazul is a widely cultivated genotype in Salamanca with good adaptability to the climate of Castilla y León (Spain) [58]. It is also characterized by high total grain protein content [59,60,61] and glutenin and gliadin composition for wheat bread making quality [62]. In this study, Gazul was cultivated in 2-L pots under water deficit conditions with the aim of achieving a better understanding of the plant’s physiological and biochemical performance in a scenario of climate change. Two water deficit regimes (LWD and TWD) were chosen to reproduce two possible scenarios. In the natural environment, prolonged drought conditions beginning at the vegetative growth stage can alter the process of storing nutrients, thereby affecting the remaining plant life cycle, while terminal drought conditions at the reproductive stage can disrupt the process of grain number formation and grain filling, thereby eventually diminishing grain yield [63].

The combination of elevated CO_2_ and high temperature with LWD treatment had a greater effect on the biomass and morphological features of flag leaves than those of TWD-treated plants (Figure 2). This was attributed to the fact that the two water deficit regimes had different detrimental effects on the accumulation of metabolites, with a key role in plant growth, or, alternatively, that they had different impacts on essential biological processes including photosynthesis or antioxidant defense response. The loss in flag leaf biomass and green area in LWD-treated plants occurred in parallel with a decrease in the total content of AsA, while the flag leaf biomass, area and the total content of AsA in the TWD-treated plants did not show significant changes in comparison with control plants (Figure 1, Figure 2 and Figure 4). The lower total content of AsA found in LWD-treated plants could account for the reduced flag leaf biomass, supported by the positive correlation (0.71) observed between both traits in the flag leaf (Appendix A). Low AsA content is known to be associated with both lower biomass and slower growth under water stress conditions [64,65] due to its role in controlling cell division and expansion [66]. Intriguingly, the flag leaves of LWD-treated plants exhibited a higher DMA than that of TWD-treated and control plants. They were shorter and thicker, a morphological change in leaves that is recognized as an indicator of drought tolerance mechanism by which the loss of water is diminished [67,68]. Regarding the effect on ears, both water deficit regimes affected the ear biomass; however, the loss in biomass was more moderate than the one observed in the flag leaves of LWD-treated plants. The fact that there were no statistically significant differences in the total content of AsA in the ears, and that AsA was not correlated with MC% (−0.06) (Appendix A), points out that this organ had a better ability to keep water moisture under water deficit conditions, as confirmed by the positive association of the LWD- and TWD-treated plants with MC%, but negative association with DMA (Figure 1E,F). This proposal finds support from other studies in which ears proved to have better drought-tolerance traits than flag leaves [69,70]. The total content of GSH did not change in the flag leaves of LWD-treated plants, implying that it could not counterbalance the negative effect that the decrease in the total content of AsA had on the flag leaf development. On the other hand, the two water deficit regimes induced a decrease in the total content of GSH in ears, but the loss in biomass was most likely buffered by both the unchanged total content of AsA among treatments and the better ability of ears to retain water. GSH is important for general plant growth and confer drought stress tolerance to plants enhancing the organ content of hormones, such as ABA, for root expansion and stomatal closure [71]. It is worth underlining that Gazul did not show dynamics changes, such as leaf rolling, during the experiment that could suggest an increase in the total content of GSH in flag leaves as a part of a protective mechanism to avoid water loss as observed in other wheat genotypes [72]. Therefore, the absence of significant changes in the total content of GSH and high *g*_s_ under water deficit conditions (Table 1) in flag leaves let us conclude that Gazul is a well-adapted genotype to drought in combination with elevated CO_2_ and high temperature.

Drought stress usually impairs the photosynthetic apparatus and leads to an over-reduction of the embedded PSII and PSI complexes, which eventually causes ROS generation [66,73]. As a result of the (photo)oxidative damage to thylakoids, the Chl content decreases and the photosynthesis performance declines [74,75]. The Chl content in the flag leaves of Gazul was very similar, regardless of the water deficit treatment (Figure 3A), implying that the oxidative stress to which the plants were exposed had to be low or moderate, or at least not high enough to have a detrimental effect on chloroplasts. In fact, *A* was measured to confirm the performance of the photosynthetic apparatus of Gazul and the results showed that *A* in the flag leaves was unexpectedly higher in LWD- and TWD-treated plants than in control plants (Figure 3B). The results were better understood when the photosynthetic parameters *V_cmax_* and *J_max_* were determined after the non-linear fitting of the *A*/*C_i_* curves (Table 1). Both *V_cmax_* and *J_max_* were higher with statistical significance in LWD- and TWD-treated plants when compared to control plants. In this regard, Gazul shows to improve both leaf nitrogen content and nitrogen use efficiency toward the photosynthetic apparatus under water deficit conditions. In addition, *g_s_* was higher under both water deficit regimes, although no statistical significance was observed when compared to control conditions (Table 1). The results here presented were not in line with those obtained from other studies in which the combination of elevated CO_2_, high temperature and moderate drought stress decreased significantly *g*_s_ and the capacity of leaf cooling of two wheat cultivars regardless of being sensitive or tolerant to drought [55]. In contrast to the former two wheat cultivars, Gazul was able to maintain high *A*, *E* and *g*_s_, a behavior that is observed in wheat cultivars with temporal reproductive drought-tolerance and eventual better grain yield [76] and also in new Australian wheat varieties with anisohydric behavior with better adaptation to early seasonal drought [77]. This let us consider Gazul as a drought tolerant genotype with anisohydric behavior and capacity to modulate both *A* and *E* so as to maintain optimal WUE at the reproductive stage (Figure 3C) and presumably leaf temperature.

Flag leaves of plants with anisohydric behavior accumulate low-molecular-weight compatible solutes, such as amino acids and soluble carbohydrates, under drought stress to maintain *g*_s_ and optimal photosynthetic activity through osmotic adjustment [76,78]. Pro plays a role as an osmoprotectant in the stabilization of protein and membrane structures and Pro accumulation is well established to improve the plant tolerance to abiotic stresses, including drought stress [79,80]. Accumulation of Pro was only observed in the flag leaves of the LWD-treated plants, whereas the Pro content increased in the ears of both water deficit treatments. To rationalize this observation, it was proposed that both LWD- and TWD-treatments affected more prominently the developing organs of plants, showing the individuals of both water deficit treatments different associations with the measured parameters as inferred from the MFA (Figure 1C,F, see below). In addition, accumulation of soluble carbohydrates (glucose, fructose, sucrose and fructan) and other amino acids playing a role in osmoprotection was previously reported in the flag leaves of LWD-treated plants [81].

Our analysis of the enzymatic antioxidant activities revealed that the activity of CAT increased moderately, with statistical significance, only in the flag leaves of LWD-treated plants, indicating that H_2_O_2_ was produced (to some extent) in excess. The activity of CAT increased where the total content of AsA diminished with statistical significance (Figure 4B and Figure 5B). The concomitant accumulation of Pro in the flag leaves of LWD-treated also pointed to an increase in ROS generation. Pro is a ROS scavenger that prevents damage to cellular components by ROS—such as hydroxyl radical formed during H_2_O_2_ decomposition within the Fenton reaction—and maintains a low NADPH/NADP^+^ ratio in response to stress [82,83]. Although APX is also an enzymatic scavenger of H_2_O_2_ with a well-established role in the protection of chloroplasts, the data showed no evidence for the enhancement of the APX activity with statistical significance under either of the two water deficit regimes (Figure 6A). GR activity of the flag leaf was the only enzyme belonging to the AsA-GSH cycle showing a statistically significant enhancement under the LWD treatment. The increase in GR activity confers a drought tolerance to the plant by recycling both GSH and AsA as a part of AsA-GSH cycle [65,84]. The scavenging of H_2_O_2_ can go through disproportionation by CAT, reduction through either APX (in which the reducing equivalents come from AsA) or AsA-independent GSH peroxidation (in which the reducing equivalents come from GSH), although the last two H_2_O_2_ decomposition pathways ultimately depend on GR [85]. Based on the results here presented, it is proposed that, under the conditions in which the content of AsA decreases as the result of moderate water deficit, both CAT and GR are the main enzymatic antioxidant actors of the H_2_O_2_ detoxification network (in conjunction with Pro), a proposal supported by the MFA (Figures S1C and 1B) and the correlogram (Appendix A). In addition, the results indicated that the TWD treatment did not severely stimulate the antioxidant defense response. None of the antioxidant enzymes belonging to the AsA-GSH cycle showed enhanced activity in the flag leaves or ears. In the TWD treatment, there was only evidence for an increase in the SOD activity with statistical significance that went in parallel with the increase in the content of Pro in the ears (Figure 4A and Figure 5A). The fact that the two water deficit regimes did not affect either the Chl content or the photosynthesis activity was here attributed to the ROS relieving effect of elevated CO_2_ in chloroplasts that hinders the oxygenase activity of Rubisco, while at the same time, diminishes H_2_O_2_ production in peroxisomes through the photorespiratory pathway [49,86]. It is thus plausible that the enhanced antioxidant defense response in the flag leaves of LWD-treated plants might mainly occur out of chloroplasts.

The results of the MFA analysis disclosed different behavior between organs under the two water deficit regimes (Figure 1). Firstly, on comparing the MFA results for flag leaves and ears separately, it was observed that the variables that had the largest contribution in the flag leaf were Area, DMA and AsA content; whereas the ones that had the largest contribution in the ears were DMA, GSH content and MC%. And secondly, the MFA analysis also revealed that the most prominent difference in organ performance depended on the water deficit regime applied. The flag leaf was mostly affected by the LWD, causing a loss of Area, MC% and AsA content and an increase in DMA. The individuals of the group of the LWD treatment were separated from the other two groups (Figure 1C). In contrast, the two water deficit regimes similarly affected the ears and the individuals of LWD and TWD treatments were grouped together (Figure 1F). In both water deficit treatments, the DMA and GSH content decreased (*r_s_* = 0.52, Appendix A), but MC% increased, in the ears. On the whole, these results agree with those of other studies, where the performance of ears and leaves of several durum wheat cultivars were investigated under water stress [70]. The former authors concluded, based on gene expression and physiological analyses, that water stress affected negatively both organs, but the effect was less detrimental in ears. Ears maintain their water and nitrogen status better than flag leaves due to their organ structure combined with better osmotic adjustment and photosynthesis performance under water deficit [70,87,88]. The higher MC% and the so-called xeromorphic anatomy of the ears confer them a better drought tolerance [89]. Although both water deficit treatments affected the ear biomass in a similar way, and the individuals of both groups were gathered mainly due to their high association with MC%, the TWD treatment (in comparison with the LWD treatment) affected the ears in a slightly different way based on the association found with the content of Pro (Figure 1E).

Regardless of the water deficit regime applied, it was observed that the antioxidant activities of some enzymes of the AsA-GSH cycle (i.e., APX and DHAR), together with the SOD, were higher in the ears when compared to the flag leaves (See Appendix A). The enhanced basal antioxidant activity in ears could thus be essential to protect cellular structures from ROS produced during the respiratory metabolism or photosynthesis [47]. This viewpoint finds support from studies where the respiration rate of the ear was observed to increase remarkably under elevated CO_2_ [90] or from other studies in which photosynthesis in the ears was proposed to have an important role in re-assimilating the CO_2_ released by grain respiration during development [91].

## 4. Materials and Methods

### 4.1. Plant Material and Experimental Growth Conditions

Pot experiments were carried out with *Triticum aestivum* L. cv Gazul, a spring wheat genotype with high yield and adaptability to the climate of the Castilla y León region (Spain). The seeds were sown in 2-L pots with a mixture of perlite:vermiculite (3:1, *v*/*v*). After seedling thinning, two seedlings were kept per pot. Plants were grown in environment-controlled chambers under a photoperiod of 16 h with a photosynthetic flux density of 400 µmol m^−2^ s^−1^ and a relative humidity of 40%/60% day/night. Pure CO_2_ was injected into the growth chamber and the concentration was monitored and kept constant at 900 µmol mol^−1^. The temperature of the growth chamber was 4 °C above the current average temperatures typically experienced throughout the life cycle of spring wheat in the province of Salamanca (Spain) and was maintained at 26 °C/16 °C day/night. The pots were irrigated with a nutrient solution containing 6 mM KNO_3_ and 2 mM Ca(NO_3_)_2_ as nitrate source twice per week and with water once per week. The concentration of the rest macro- and micronutrients in the nutrient solution is described in [22]. Two different water deficit regimes were established: (i) LWD with 65% field capacity beginning at the vegetative growth stage (Zadoks stage Z.14 [92]) and (ii) TWD with 50% field capacity beginning at the ear emergence stage (Z.59). Control plants were watered at full field capacity. The field capacity values were the percentage of the total weight of control pots at full field capacity after mixing dry soil with abundant water, allowing the mixture to drain by gravity for 24 h.

### 4.2. Photosynthetic Parameter Determination

The gas exchange measurements, i.e., *A*, *g*_s_, *C_i_*, *V_cmax_*, *J_max_*, *E*, *R*_d_ and WUE, were obtained from flag leaves of the main stem of the wheat plants using a portable photosynthesis system CIRAS-3 (PP Systems, Amesbury, MA). The values for the photosynthetic parameters, i.e., *V_cmax_*, *J_max_* and *R_d_*, were obtained after non-linear fitting of the *A*/*C_i_* curves using the LeafWeb service [93]. The CIRAS-3 parameter settings were as follows: temperature in cuvette, 25 °C; light intensity, 1500 µmol m^−2^ s^−1^ with a blue:red LED ratio of 10:90; cuvette flow of air, 250 cm^3^ min^−1^; relative humidity, 50%; and a stepwise gradient of CO_2_ from 60–2000 ppm in 30 min with 70 s of acclimation per level and two points per level, as reported in [17]. Before the CO_2_ gradient was initiated, the attached flag leaf was pressed between the upper and lower gaskets of the leaf cuvette head and pre-acclimated for 20 min. Gas exchange measurements were performed at the anthesis growth stage (Z.69).

The Chl content in the attached flag leaves was measured using a non-destructive Dualex instrument (FORCE-A, Paris, France) as described in [94].

### 4.3. Biomass Parameters

FW of the aerial plant organs (flag leaves and ears) were determined seven days after anthesis. After drying them at 60 °C for 48 h in an oven, each separated green organ was weighed to determine DW. MC% was calculated as follows:(1)$$\mathrm{MC}\%=\frac{\mathrm{FW}-\mathrm{DW}}{\mathrm{FW}}\times 100,$$

The area per organ unit was measured using an electronic planimeter (Li-3050A, Lincoln, ME, USA), and DMA was then calculated.

### 4.4. Sampling Procedure

For each of four biological replicates, the flag leaves and ears of the three oldest tillers of two plants per treatment were harvested seven days after anthesis and immediately submerged in liquid N_2_ and preserved at −80 °C until processing. The frozen plant material was ground using a pre-cooled mortar and pestle with liquid N_2_ and the fine powder transferred into microfuge tubes and kept at −80 °C until enzymatic antioxidant activity assay and metabolite content analysis were performed (see below).

A 96-well microplate reader FLUOstart^®^ Omega (BMG Labtech, Ostenberg, Germany) was used for all the spectrophotometric methods described in the following sections.

### 4.5. Proline Content Quantification

Pro content was measured following the protocol described in [95]. Briefly, an amount of 40−45 mg of the frozen plant material was mixed with a volume of 0.5 mL of the extracting buffer containing 3% (*w*/*v*) sulfosalicylic acid. The mixture was manually homogenized and centrifuged twice for 10 min at 13,000× *g* at 10 °C An aliquot of 250 µL of each supernatant was mixed with 250 µL of glacial acetic acid and 500 µL of 2.5% (*w*/*v*) ninhydrin, glacial acetic acid. The reaction mixture was heated at 99 °C for 40 min in a digital dry bath (Bio-Rad, Hercules, CA, USA). The solution was centrifuged at 13,000× *g* for 5 min at 10 °C. A volume of 800 µL of the supernatant was mixed with an equal volume of toluene to transfer the Pro-ninhydrin chromophore into the organic phase. The concentration of the Pro-ninhydrin chromophore was determined at 520 nm. A standard curve was prepared using commercial L-Pro (Merck, Kenilworth, NJ, USA).

### 4.6. Ascorbate and Glutathione Content Quantification

A weighed amount of about 40−45 mg of fine powder of the plant material was mixed with 0.8 mL of 3% (*w*/*v*) HCl, 1 mM Na_2_EDTA (extraction buffer) in microfuge tubes containing approximately 100 mg of 0.75−1.0 mm glass grinding beads to measure both the total content of AsA (AsA plus DHA) and GSH (GSH plus GSSG). The sample mixtures were vigorously shaken in a Mini-Beadbeater-16 homogenizer (BioSpec, Bartlesville, OK, USA) for 1 min and then centrifuged for 15 min at 13,200× *g* at 4 °C. The supernatant was kept at 4 °C.

The total content of AsA was measured following the protocol described in [96,97] with some modifications. A volume of 100 µL of the acidic supernatants was mixed (in the following order) with an equal volume of the extraction buffer, 600 µL of H_2_O, 200 µL of 1 M sodium succinate and 0.5 M KOH. A volume of 300 µL was deposited in each well and the absorbance values at 265 nm were directly recorded with no further additions. A volume of 3 µL of 300 mM dithiothreitol (DTT) was added to the wells in order to measured DHA. AsA was determined by adding 3 µL of ascorbate oxidase (AO) from a 100 U/mL stock solution to the wells. Changes in absorbance at 265 nm were monitored for 12 min until reaching steady values. The total content of AsA was determined from the difference between the absorbance of wells containing DTT and AO. An extinction coefficient of 13,250 M^−1^cm^−1^ at 265 nm for (reduced) AsA was determined in the reaction mixture at pH 5.18. The results were expressed as µmol g FW^−1^.

The total content of GSH was determined following the enzymatic-recycling procedure described in [98]. The assay was adapted to a 96-well microplate reader by Pérez-López et al. [97] with some modifications. A volume of 100 µL of the acidic supernatants were mixed (in the following order) with 100 µL of the extraction buffer, 350 µL of 1 M succinate and 0.5 M KOH, and 350 µL of 6.3 mM Na_2_EDTA, 0.32 mM NADPH, 250 mM Tris-HCl, pH 7.0. A volume of 270 µL of the neutralized reaction mixtures was transferred into the microplate wells already containing 3 µL of a stock solution of 100 U/mL GR. The measure started by adding 30 µL of 6 mM DTNB. The reduction of DTNB was monitored at 412 nm. The total content of GSH was then evaluated by comparing the results with a standard curve with GSH. The results were expressed as µmol g FW^−1^.

### 4.7. Enzymatic Antioxidant Activity Assays

All the extraction procedures for the enzyme assays were performed following the steps described in [99] with modifications. A volume of 0.5 mL of the extraction buffer consisting of 0.1 Na_2_EDTA, 0.2% *v*/*v* Triton X-100, 0.1 mM phenyl-methylsulfonyl fluoride (PMSF), 2 mM DTT and 50 mM Tris-HCl pH 7.8 was mixed with 40 mg of the frozen plant material in microfuge tubes containing approximately 100 mg of 0.75−1.0 mm glass grinding beads to measure the following enzyme activities: CAT (EC 1.11.1.6), SOD (EC 1.15.1.1) and GR (EC 1.6.4.2). The sample mixtures were vigorously shaken in a Mini-Beadbeater-16 homogenizer (BioSpec, Bartlesville, OK, USA) for 1 min and then centrifuged for 15 min at 16,100× *g* at 4 °C. A volume of 180 µL of the supernatants was gel-filtered through Shepadex G-25 columns previously equilibrated with 0.1 EDTA, 0.2% *v*/*v* Triton X-100, 50 mM Tris-HCl pH 7.8.

The CAT activity was measured following the Aebi procedure [100] with some modifications [99,101]. The reaction began after adding 25 mM H_2_O_2_ to the assay mixture. The consumption of H_2_O_2_ was monitored at 240 nm at 25 °C in a buffer consisting of 50 mM KH_2_PO_4_/K_2_HPO_4_ pH 7.0.

The SOD activity was determined following the reaction competition between SOD and ferricytochrome *c* (cyt *c*) for the O_2_^●−^ produced enzymatically by xanthine oxidase [99,102]. The measurement was performed at 25 °C in a reaction buffer containing 0.1 mM Na_2_EDTA, 0.82 mM xanthine, 0.014 mM cyt *c*, 50 mM KH_2_PO_4_/K_2_HPO_4_ pH 7.8. Changes in absorbance at 550 nm upon cyt *c* reduction by O_2_^●−^ at a rate of 1.5 units h^−1^ were adjusted by trial and error for the sample blank using different amounts of xanthine oxidase units. Aliquots of the sample supernatants ranging between 3−7 µL were used to reach an inhibition of 50% of cyt *c* reduction. The reaction buffer was added to the well until completing a final volume of 300 µL. One unit of SOD activity was defined as the amount of enzyme in the plant samples that inhibited the rate of the cyt *c* reduction by 50%.

The protocol described in [103] and modified by Perez-López et al. [99] was used to measure the GR activity. The reaction was carried out at 25 °C and the absorbance decrease due to NADPH oxidation was monitored at 340 nm. A volume of 40 µL of the extracted samples was loaded in each well together with 255 µL of the reaction mixture containing 1 mM Na_2_EDTA, 2 mM MgCl_2_, 0.5 mM GSSG, 50 mM HEPES pH 7.8. The reaction began after the addition of 15 µL of 15 mM NADPH. An extinction coefficient of 6220 M^−1^cm^−1^ at 340 nm for NADPH was determined in the reaction mixture at pH 7.8.

The extraction for the measure of APX (EC 1.11.1.11), DHAR (EC 1.8.5.1) and MDHAR (EC 1.6.5.4) was carried out in a buffer consisting of 0.1 mM Na_2_EDTA, 0.2% (*v*/*v*) Triton X-100, 5 mM cysteine-HCl, 0.1 mM PMSF, 1% polyvinylpolypyrrolidone, 50 mM KH_2_PO_4_/K_2_HPO_4_ pH 7.8. Particularly, the extraction buffer for the APX activity assay also contained 2 mM AsA to avoid inactivation of this enzyme during the activity assay. An amount of 40 mg of the frozen plant material was mixed with 0.5 mL of the extraction buffer. The extraction mixtures were homogenized and centrifuged as described above.

The APX activity was determined by measuring the oxidation rate of AsA at 290 nm [99,104]. The decrease in the AsA concentration was monitored in the reaction buffer containing 0.8 mM AsA, 1.2 mM H_2_O_2_, 50 mM HEPES-NaOH pH 7.6. An extinction coefficient of 2745 M^−1^cm^−1^ at 290 nm for AsA was determined in the reaction mixture at pH 7.6.

In order to measure the DHAR activity, the AsA formation via DHA reduction was followed according to the method described in [105] with modifications [99]. The increase in AsA concentration was determined in the reaction buffer 0.1 mM Na_2_EDTA, 2.5 mM GSH, 50 mM KH_2_PO4/K_2_HPO4 pH 6.6 after adding 10 µL of 0.2 mM DHA. An extinction coefficient of 14,700 mM^−1^cm^−1^ at 265 nm for AsA was determined in the reaction mixture at pH 6.6.

The MDHAR activity assay was performed according to the method described in [106] with modifications [99]. The NADH oxidation by MDHAR was monitored following the decrease in absorbance at 340 nm. An aliquot of 10 µL of supernatant was mixed with 288 µL of a buffer containing 2 mM AsA, 0.2 mM NADH, 50 mM KH_2_PO_4_/K_2_HPO_4_ pH 7.2. The reaction began after the addition of 1 unit of AO (Merck, Kenilworth, NJ). An extinction coefficient of 6220 M^−1^cm^−1^ at 340 nm for NADH was determined in the reaction mixture at pH 7.2.

The protein concentration for each of the enzyme activity extracts was determined according to the Bradford method [107] with the modifications described in [108]. The bovine serum albumin was used as the standard protein.

### 4.8. Statistical Analysis

Randomization of replicates was applied to the experiment design. Four replicates (*n* = 4) per water regime and control conditions were used. The obtained experimental data were analyzed using the R statistical software [109]. MFA was carried out to explore the interactive effect of organ and water deficit treatments on the measured parameters, except for photosynthetic parameters, that were only measured in flag leaves [110,111]. Correlograms based on Sperman’s correlation were performed with the same parameters in MFA analysis using all dataset for both organs and then splitting up between the flag leaf data and the ear data [112,113]. One-way ANOVA analyses and Tukey post-hoc tests were independently performed for ears and flag leaves to assess the effect of the water deficit treatments on biomass, photosynthesis (only in flag leaves), metabolite content and enzymatic antioxidant activities. Previously, data normality and homoscedasticity were tested by Shapiro and Levene test, respectively. Krustal-Wallis variation of ANOVA and pairwise-Wilcoxon post-hoc test were performed in cases where an assumption of normality was not achieved. ANOVA with heteroscedasticity correction and pairwise-t post-hoc test was conducted in which data had normality but no homogeneity of variance. Two group comparison tests (T-test) were also applied for the comparison of the AsA-GSH cycle enzyme activities between the flag leaves and the ear. Wilcoxon tests were performed when the data lacked either normality or homogeneity of variance. The analyses used for each variable are specified in Appendix A. Significant differences were considered when the *p*-value was lower than 0.05. Further details are given in [60]. Origin 2021 (OriginLab Corporation, Northampton, MA, USA) was used for drawing bar plots. The data presented in this study are available in Appendix A.

## 5. Conclusions

Our analysis of the physiological and biochemical traits of Gazul defines it as a wheat cultivar with high capacity of adaptation to water deficit conditions under controlled environments combining elevated CO_2_ and high temperature. LWD conditions reduced the flag leaf biomass of Gazul, but in contrast, the DMA increased as an adaptive response to water deficit conditions. The lower content of AsA had a dual impact on the biomass and antioxidant status of the flag leaves of LWD-treated plants. The accumulation of Pro (as an osmoprotectant and ROS scavenger) and the increase in the antioxidant activity of CAT and GR were indicative of oxidative stress in the flag leaves of LWD-treated plants. No equivalent trends in the biomass or antioxidant system were observed in the flag leaves of TWD-treated samples. The absence of a deleterious effect on the flag leaf Chl content, accompanied by reduced leaf biomass and high *A* and *g*_s_, suggests that Gazul followed an anisohydric strategy at the reproductive stage that could be essential for the attenuation of grain filling losses under drought conditions. The higher MC% of the ears under water deficit and the enhanced basal antioxidant activity that we observed are in agreement with the findings of other studies, supporting the greater drought tolerance capacity of ears. The search for this type of physiological and biochemical traits of ears, remarkably different from those of flag leaves, is of relevance for breeding programs aiming to improve grain yield under future climate change scenarios.

## Figures and Tables

**Figure 1 plants-11-02384-f001:**
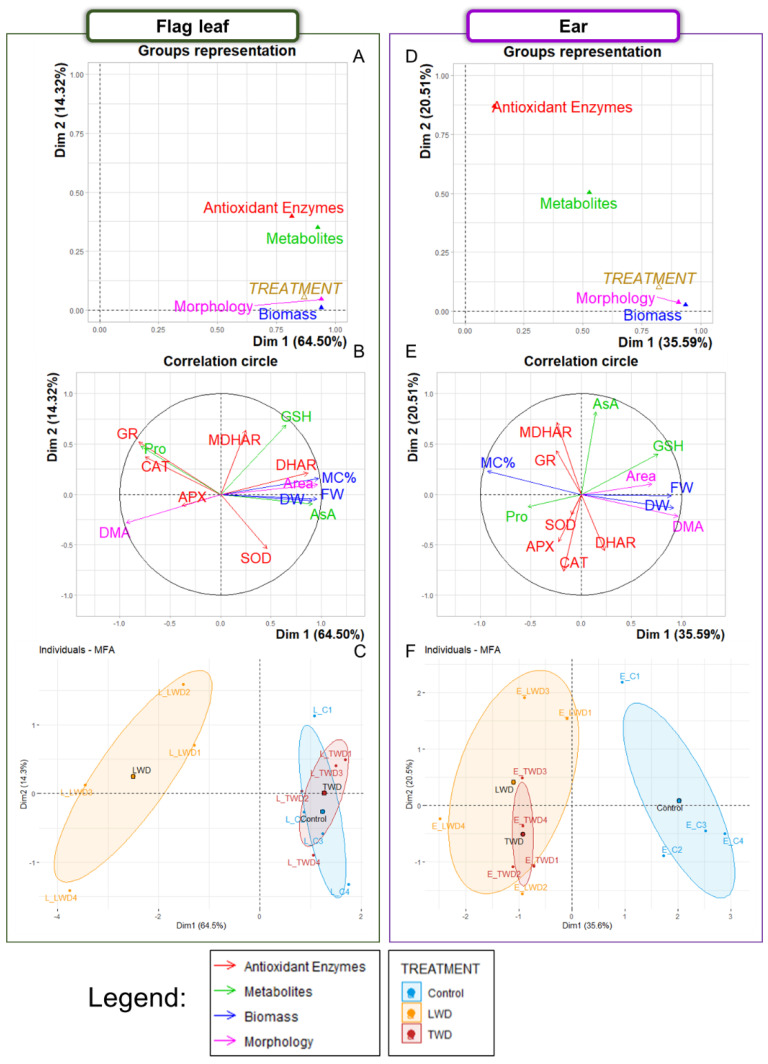
Multiple factorial analysis for flag leaves (Panels **A**–**C**) and ears (Panels **D**–**F**) of potted wheat subjected to two different water regimes under elevated CO_2_ and high temperature. Dim, Dimension; DW, dry weight; FW, fresh weight; MC%, moisture content percentage; Pro, proline content; AsA, total content of ascorbate (AsA plus DHA); GSH, total content of glutathione (GSH plus GSSG); SOD, superoxide dismutase; CAT, catalase; APX, ascorbate peroxidase; DHAR, dehydroascorbate reductase; MDHAR, monodehydroascorbate reductase; GR, glutathione reductase. Two factors were defined: *ORGAN*, which included the flag leaf and ear categories, and *TREATMENT*, which included control, long water deficit (LWD) and terminal water deficit (TWD) categories. The green biomass and biochemical traits were divided into four groups: *Antioxidant Enzymes* (CAT, SOD, APX, DHAR, MDHAR and GR), *Biomass* (FW, DW and MC%), *Metabolites* (Pro, AsA and GSH), *Morphology* (Area and DMA). Panels A and D contain the group representation plots showing the correlation between the trait groups of the continuous variables and the two factors with the first and second dimensions of the MFA axes. Triangles represent the relative position of each group with the axes. Panel B and E contain the correlation circle plots showing the correlation of variables with the MFA axes. Panel C and F contain the individuals-MFA plots showing the position of the individuals in the MFA by organ and treatment. Dots indicate individuals. Ellipses around each treatment were added.

**Figure 2 plants-11-02384-f002:**
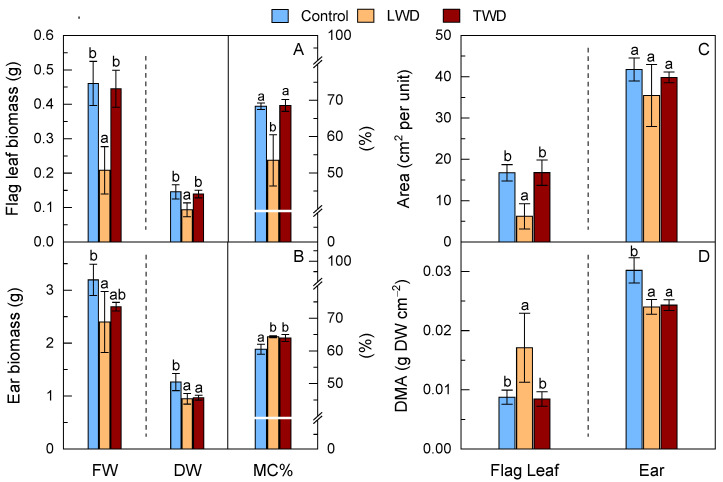
Effect of two water deficit treatments under combined elevated CO_2_ and high temperature on the fresh weight (FW), dry weight (DW) biomass and moisture content percentage (MC%) of flag leaf (Panel **A**) and ear (Panel **B**), green area per organ unit (Panel **C**) and dry matter per unit of area (DMA) (Panel **D**) of potted wheat plants. Means of data from control (blue bars), long water deficit (LWD, orange bars) and terminal water deficit (TWD, dark red bars) are plotted together with the standard deviation (SD). Different letters above the bars (a and b) indicate statistically significant differences among treatments for each organ after post-hoc test analyses (*p*-value ≤ 0.05, *n* = 4, see Appendix A).

**Figure 3 plants-11-02384-f003:**
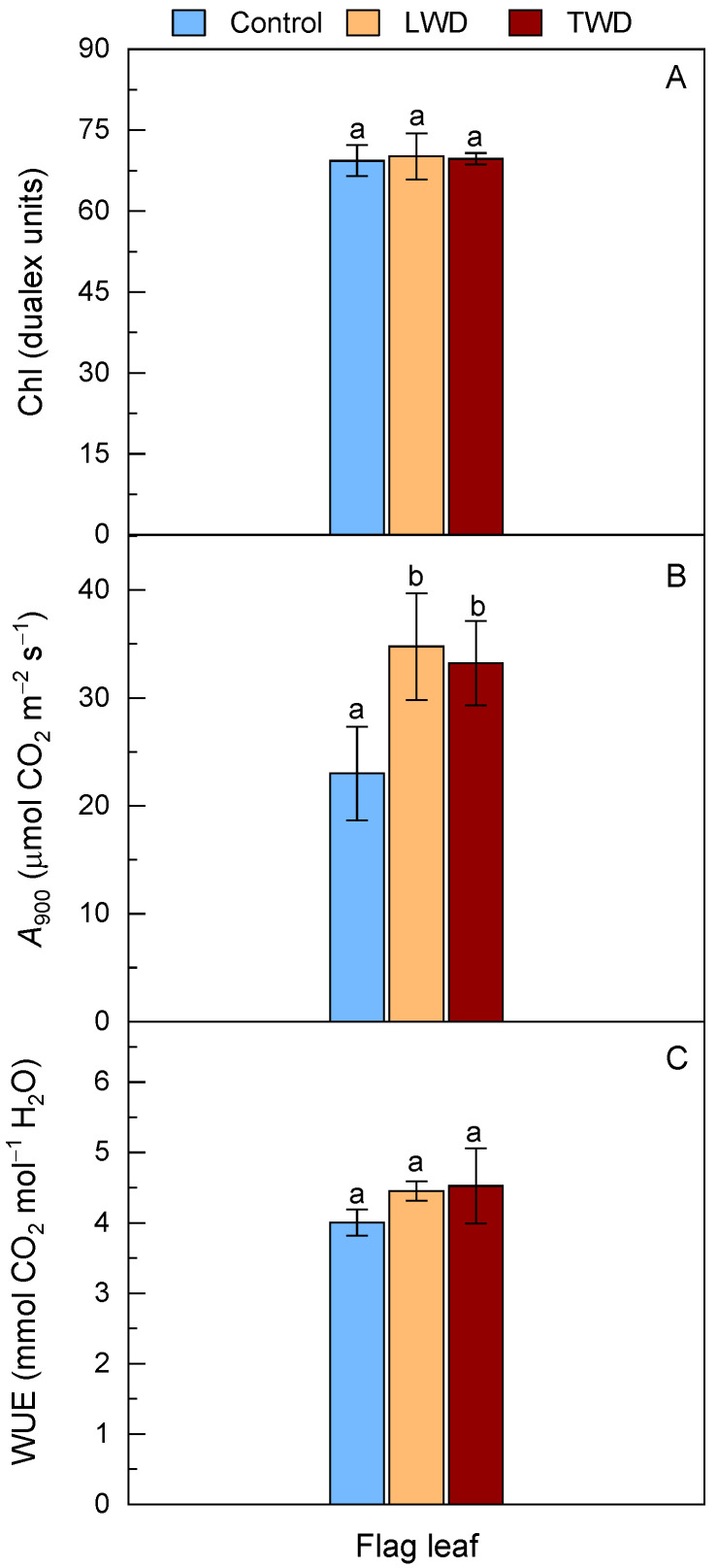
Effect of two water deficit treatments under combined elevated CO_2_ and high temperature on chlorophyll (Chl) (Panel **A**), leaf net CO_2_ assimilation rate (*A*) (Panel **B**) and water use efficiency (WUE) (Panel **C**) of the flag leaves of potted wheat plants. Leaf gas exchange was measured at 900 ppm of CO_2_. Legend is as described in Figure 2. Bars represent mean values ± SD. Different letters above the bars (a and b) indicate statistically significant differences among treatments for each organ after post-hoc test analyses (*p*-value ≤ 0.05, *n* = 4, see Appendix A).

**Figure 4 plants-11-02384-f004:**
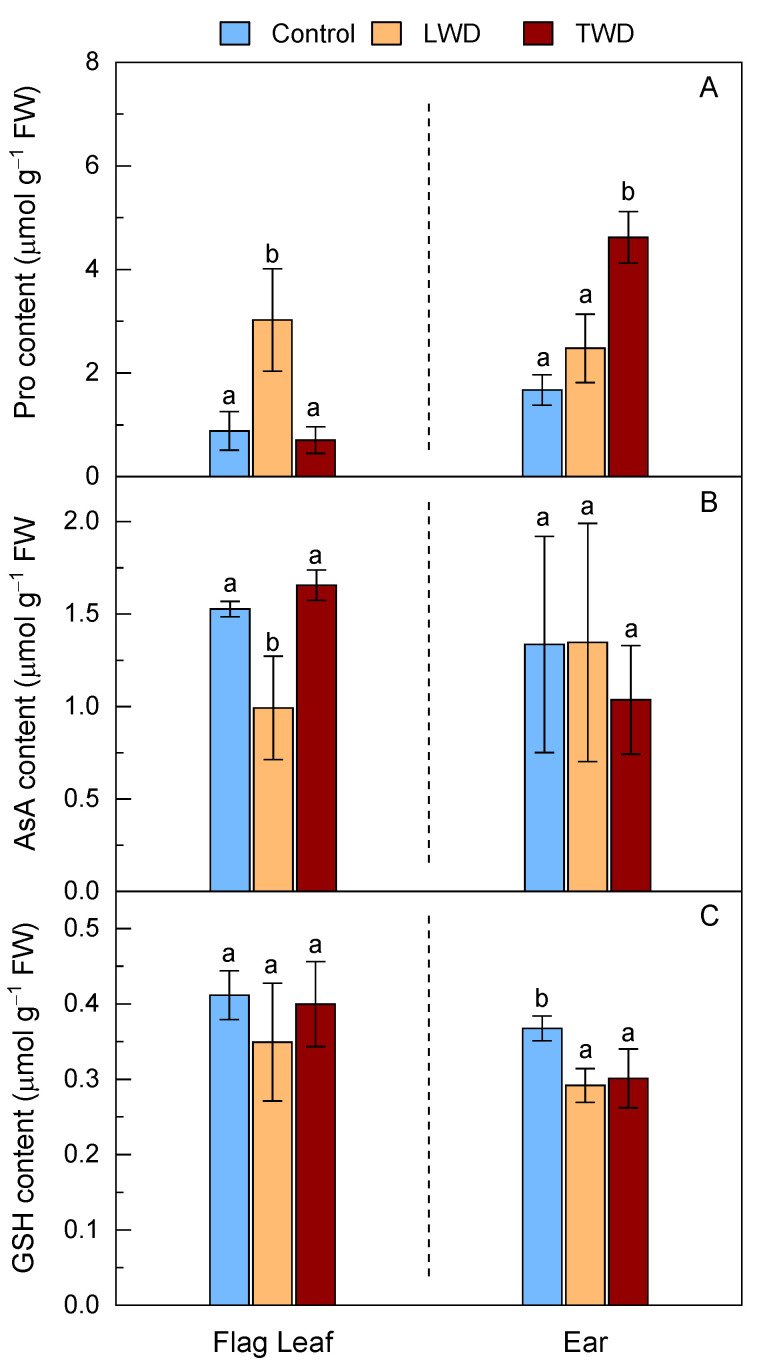
Effect of two water deficit treatments under combined elevated CO_2_ and high temperature on Pro (Panel **A**), total content of ascorbate (AsA plus DHA) (Panel **B**) and total content of glutathione (GSH plus GSSG) (Panel **C**) in the flag leaves and ears of potted wheat. Legend is as described in Figure 2. Bars represent mean values ± SD. Different letters above the bars (a and b) indicate statistically significant differences among treatments for each organ after post-hoc test analyses (*p*-value ≤ 0.05, *n* = 4, see Appendix A).

**Figure 5 plants-11-02384-f005:**
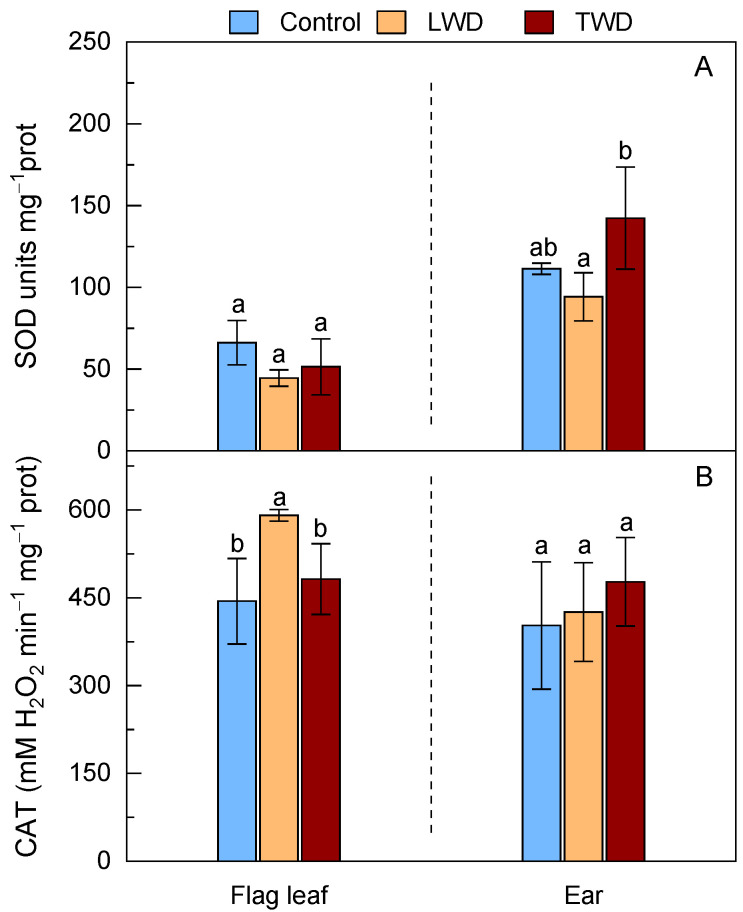
Effect of two water deficit treatments under combined elevated CO_2_ and high temperature on superoxide dismutase (SOD) (Panel **A**) and catalase (CAT) (Panel **B**) activities in the flag leaves and ears of potted wheat. Legend is as described in Figure 2. Bars represent mean values ± SD. Different letters above the bars (a and b) indicate statistically significant differences among treatments for each organ after post-hoc test analyses (*p*-value ≤ 0.05, *n* = 4, see Appendix A).

**Figure 6 plants-11-02384-f006:**
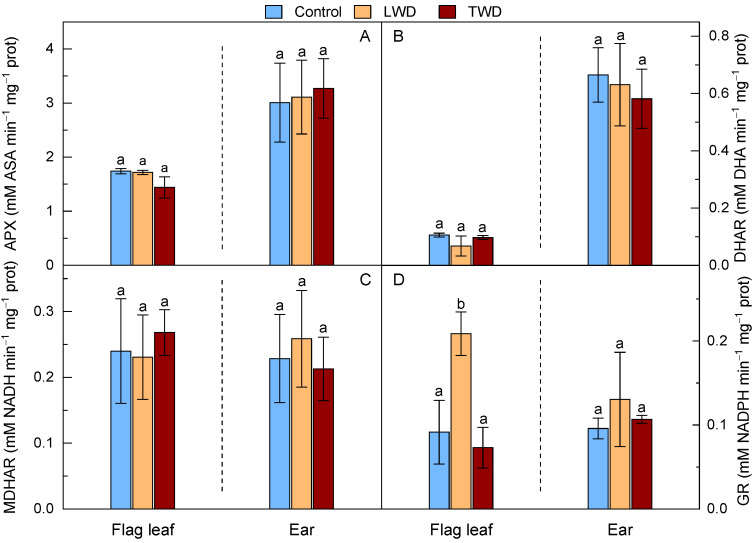
Effect of two water deficit treatments under combined elevated CO_2_ and high temperature on ascorbate peroxidase (APX) (Panel **A**), dehydroascorbate reductase (DHAR) (Panel **B**), monodehydroascorbate reductase (MDHAR) (Panel **C**) and glutathione reductase (GR) (Panel **D**) activities in flag leaf and ear of potted wheat. Legend is as described in Figure 2. Bars represent mean values ± SD. Different letters above the bars (a and b) indicate statistically significant differences among treatments for each organ after post-hoc test analyses (*p*-value ≤ 0.05, *n* = 4, see Appendix A).

**Table 1 plants-11-02384-t001:** Effect of long and terminal water regimes (LWD and TWD) on the photosynthetic and gas exchange parameters of the flag leaf of wheat growing under combined elevated CO_2_ and high temperature. Means of data are represented ± SD. Different letters next to values (a and b) indicate statistically significant differences among treatments after post-hoc test analyses (*p*-value ≤ 0.05, *n* = 4, see Appendix A). The number 900 stands for the value for the CO_2_ concentration in ppm at which the photosynthetic parameters were determined.

Treatment	*V_cmax_* * (µmol CO_2_ m^−2^s^−1^)	*J_max_* (µmol e^−^ m^−2^s^−1^)	*R**_d_* (µmol CO_2_ m^−2^s^−1^)	*g_s_*_900_ (mol H_2_O m^−2^s^−1^)	*E*_900_ (mmol H_2_O m^−2^s^−1^)	*C_i_*_900_ (ppm CO_2_)
Control	56.85 ± 9.14b	128.95 ± 16.61b	2.22 ± 0.49a	0.35 ± 0.18a	5.31 ± 20a	709.75 ± 61.95a
LWD	78.73 ± 9.82a	181.78 ± 27.73a	1.3 ± 0.27b	0.58 ± 0.13a	7.35 ± 0.76a	714.38 ± 23.94a
TWD	81.66 ± 8.22a	176.55 ± 0.62a	1.24 ± 0.15b	0.6 ± 0.06a	7.35 ± 0.45a	747.63± 16.38a

* *V_cmax_*, maximum carboxylation rate by Rubisco; *J_max_* maximum electron transport rate; *R_d_*, respiration rate in the light; *g_s_*, stomatal conductance to water vapor; *E*, transpiration rate and *C_i_*, sub-stomatal (intercellular) CO_2_ concentration.

## Data Availability

Not applicable.

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
