# Peer review of "Physiological and Antioxidant Response to Different Water Deficit Regimes of Flag Leaves and Ears of Wheat Grown under Combined Elevated CO2 and High Temperature"

_plants, 2022, doi:10.3390/plants11182384_

Round 1

Reviewer 1 Report

1. Genetic correlation estimates between variables were imprecisely descripted in the MFA. Please generate correlograms and conduct descriptive statistics include broad sense of heritability to better interpret this dataset.

2. The description for multiple comparison using one-way ANOVA was vague. Please detailed describe the algorithm such as Tukey etc., that was used in the method section.

Author Response

REVIEWER 1

We would like to thank Reviewer 1 for their critical reading. We have followed their comments to improve the manuscript, particularly the sections of Materials and Methods, and Results. Because of this, new supplementary figures showing the correlograms of the continuous variables are presented to further reinforce the multifactorial analysis and our main conclusions.

1. Reviewer’s comment

1. Genetic correlation estimates between variables were imprecisely descripted in the MFA. Please generate correlograms and conduct descriptive statistics include broad sense of heritability to better interpret this dataset.

1. Authors’ response

In our study, we have not performed any genetic analysis to test whether or not there was significant genetic variation between the individuals. Our experiment was performed with a single spring wheat genotype, named Gazul, characterized by being widely-distributed commercially in Castilla y León (Spain) and having high yield performance. We understand that estimation of the heritability trait does not apply in our study when using one single wheat genotype. From the beginning, we did not plan to perform any genotype x environment interaction and so the phenotypic variation was expected to be mainly attributed to the water deficit regimes applied to the potted-wheat plants. To study the phenotypic variation, we used standard statistical analysis tools as those described in the manuscript. In addition, the MFA was performed as an initial exploratory analysis with the aim of deciding how to present our main findings in the Results section. The result of the first exploratory analysis was essential to take the decision of splitting the data in two main blocks, each corresponding with one of the two organs. As suggested by the reviewer, we have now performed correlation analyses between the continuous variables. The correlograms are presented as supplementary figures and cited in the Results and Discussion sections.

2. Reviewer’s comment

2. The description for multiple comparison using one-way ANOVA was vague. Please detailed describe the algorithm such as Tukey etc., that was used in the method section.

2. Authors’ response

The Statistical Analysis subsection is now better described and includes the statistical functions used to check the assumptions of normality and homocedasticity of the data sets, to perform correlation analyses and post-hoc tests. The supplementary material (Excel file) includes all the statistical analyses and type of corrections used in the post-hoc tests.

Reviewer 2 Report

The manuscript under the title"Physiological and antioxidant response to different water deficit regimes of flag leaves and ears of wheat grown under combined elevated CO2 and high temperature" is very interesting, but my concern about the methodology should be improved, authors should revise the following notices.

- the authors should include references to support this information.

- How old is the plant? the authors should be mentioned in the experimental growth in each treatment the plant age, also where the control.

- Regarding field capacity how do the authors determine it?

- I think the methods of Proline,  Ascorbate, glutathione, and Enzymatic antioxidant activity are written in detail, so, I suggest to be short as possible as.

-

Author Response

  1. Reviewer’s comment

The manuscript under the title "Physiological and antioxidant response to different water deficit regimes of flag leaves and ears of wheat grown under combined elevated CO2 and high temperature" is very interesting, but my concern about the methodology should be improved, authors should revise the following notices.

  1. Authors’ response

We would like to thank Reviewer 2 for their very constructive and positive view of the manuscript. We have carefully paid attention to their concern about the methodology of the manuscript and we have introduced some changes in the Materials and Methods to reinforce this part.

  1. Reviewer’s comment

- the authors should include references to support this information.

  1. Authors’ response

New references have been included in the Materials and Methods section to particularly complete the description of the subsections concerning photosynthetic parameter determination and ascorbate content quantification, in which no references were included.

  1. Reviewer’s comment

- How old is the plant? the authors should be mentioned in the experimental growth in each treatment the plant age, also where the control.

  1. Authors’ response

A better description of the plant age is now included in the Materials and Methods section. To avoid ambiguity about plant age and growth stage, the decimal codes of the Zadoks growth scale was used. For example, the long water deficit (LWD) regime started at the vegetative stage, Z 14, and the terminal water deficit (TWD) regime started at the ear emergency, Z 59. Leaf gas exchange was measured at anthesis, Z 69. A sentence describing the water conditions at which control plants were grown is now included. In particular, control plants were watered at full field capacity.

  1. Reviewer’s comment

- Regarding field capacity how do the authors determine it?

  1. Authors’ response

The field capacity values at 65% and 50% for LWD and TWD regimes respectively were the percentage of the total weight of control pots at full field capacity after mixing dry soil with abundant water and allowing the mixture to drain by gravity for 24 h.

  1. Reviewer’s comment

- I think the methods of Proline,  Ascorbate, glutathione, and Enzymatic antioxidant activity are written in detail, so, I suggest to be short as possible as.

  1. Authors’ response

The Plants instructions for authors state that Plants has no restrictions on the length of manuscripts, provided that the text is concise and comprehensive. Because some of the original analytical methods were first designed to measure enzyme activities or quantify metabolites in standard 1-cm cuvettes and spectrophotometers, we found necessary to give some further details about the way in which the analytical methods were adopted to 96-well plates and how the technical replicates were placed in the 96-well plates to optimize the analysis. In addition, the ratio between the weighed amount of the plant material and the volume of the extraction or assay buffers were different to those reported in the original studies because either the plant material had a different origin, or it came from different organs of the plant. So, in our mind it was to facilitate the analytical analyses to other researchers that could be interested in reproducing the methods we describe in the article. Notwithstanding, we have revised the subsections suggested by the reviewer and we have reduced their length to avoid redundancy.

Reviewer 3 Report

Thank you for the opportunity to review this manuscript, dealing with interesting findings entitled “Physiological and antioxidant response to different water deficit regimes of flag leaves and ears of wheat grown under combined elevated CO2 and high temperature”. All findings are interesting, and the article includes a balanced and critical view of the outcomes. However, their representation of statistical analysis between the groups is confusing. I am unable to find what the author is representing by pointing to “a”, “b”, and “c”. Does it represent significance? If yes, then to which they are comparing. Also, there’s no information about these alphabets in Figure legends. It is recommended to resubmit the manuscript with proper statistical analysis and description of significance with each group. Also, it would be nice to show a scatter-bar graph instead of a bar graph only, so that readers could see the individual value of the data. The author also needs to add samples size in each figure legend.

Author Response

  1. Reviewer’s comment

Thank you for the opportunity to review this manuscript, dealing with interesting findings entitled “Physiological and antioxidant response to different water deficit regimes of flag leaves and ears of wheat grown under combined elevated CO2 and high temperature”. All findings are interesting, and the article includes a balanced and critical view of the outcomes.

  1. Authors’ response

We would like to thank Reviewer 3 for their critical reading and constructive view of the manuscript. We have improved the Materials and Methods section and the presentation of the results to reinforce the main conclusions we present in our study.

  1. Reviewer’s comment

However, their representation of statistical analysis between the groups is confusing. I am unable to find what the author is representing by pointing to “a”, “b”, and “c”. Does it represent significance? If yes, then to which they are comparing. Also, there’s no information about these alphabets in Figure legends. It is recommended to resubmit the manuscript with proper statistical analysis and description of significance with each group.

  1. Authors’ response

Reviewer 3 is correct and different letters above the bars indicate statistically significant differences at p-value < 0.05 among treatments for each organ (one-way ANOVA). The meaning of the letters is now found in the legend of Figures 2-6.

  1. Reviewer’s comment

Also, it would be nice to show a scatter-bar graph instead of a bar graph only, so that readers could see the individual value of the data. The author also needs to add samples size in each figure legend.

  1. Authors’ response

We have decided to replace the standard error of the mean value of each bar with the corresponding standard deviation (SD) in Figures 2-6 to better visualize the amount of dispersion of the individual values. We have taken this decision, instead of using symbols for each individual value as the reviewer suggested, because there are several bars throughout Figure 2-6 in which the SD is so small that the four data points clustered together, and they could not be identified from each other. A full segment representing SD is now also plotted. The number of samples per treatment is four (n = 4). This latter value is also indicated in Materials and Methods section.

Round 2

Reviewer 1 Report

I have noticed that the issues have been addressed based on the reviewer’s comments in the revised version, and three correlograms have been added in the supplement file. No further revision is required.

Reviewer 2 Report

The manuscript is ready now for publishing,

Reviewer 3 Report

N/A